# Correlations between the selection of topics by news media and scientific journals

**Melanie Leidecker-Sandmann**[1]*, **Lars Koppers**[2], **Markus Lehmkuhl**[1]

**1** Department of Science Communication, Institute of Technology Futures, Karlsruhe Institute of Technology, Karlsruhe, Germany, **2** Science Media Center Germany, Köln, Germany

* leidecker-sandmann@kit.edu

## Abstract

The aim of this study is to reveal a robust correlation between the amount of attention international journalism devotes to scientific papers and the amount of attention scientific journals devote to the respective topics. Using a Mainstream-Media-Score (MSM) $\geq 100$ (which we regard as an indicator for news media attention) from the altmetrics provider *Altmetric*, we link 983 research articles with 185,166 thematically similar articles from the *PubMed* database (which we use to operationalize attention from scientific journals). The method we use is to test whether there is a concomitant increase in scientific attention after a research article has received popular media coverage. To do so, we compare the quotient of the number of thematically similar articles published in scientific journals during the period before and after the publication of an MSM $\geq 100$ article. Our main result shows that in 59 percent of cases, more thematically similar articles were published in scientific journals after a scientific paper received noteworthy news media coverage than before ($p < 0.01$). In this context, we neither found significant differences between various types of scientific journal ($p = 0.3$) nor between scientific papers that were originally published in renowned opinion-leading journals or in less renowned, non-opinion-leading journals ($p = 0.1$). Our findings indicate a robust correlation between the choice of topics in the mass media and in research. However, our study cannot clarify whether this correlation occurs because researchers and/or scientific journals are oriented towards public relevance (publicity effect) or whether the correlation is due to the parallelism of relevance attributions in quality journalism and research (earmark hypothesis). We infer that topics of social relevance are (more) likely to be picked up by popular media as well as by scientific journals. Altogether, our study contributes new empirical findings to the relationship between topic selection in journalism and in research.

**Data Availability Statement:** The datasets generated during and/or analyzed during the current study are available from figshare repository

# 1 Introduction

Although correlations between journalism and research communication are widely considered plausible (e.g. regarding topic selection and editing [1–4]), they have only been the subject of systematic empirical research comparatively seldom. The current study aims to expand the existing canon of empirical work and investigates whether there is a robust correlation

at https://doi.org/10.6084/m9.figshare.20101811.
v1.

**Funding:** This work was supported by grant
411038189 from the German Research Foundation
(DFG), Bonn within the framework program
"Medializing brain diseases: interactions between
research and mass media" (https://gepris.dfg.de/
gepris/projekt/411038189?language=en) and by
grant 94838 from the Volkswagen Foundation
within the framework program "Entwicklung von
Methoden und Tools für eine datengestützte
Wissenschaftskommunikation" ["Development of
methods and tools for data-based science
communication", Hannover (https://www.
volkswagenstiftung.de/). The author who received
both awards was M. L. The funders had no role in
study design, data collection and analysis, decision
to publish, or preparation of the manuscript. There
was no additional external funding received for this
study. We acknowledge support by the Open
Access Publication Fund of the Karlsruhe Institute
of Technology.

**Competing interests:** The authors have declared
that no competing interests exist.

between topics selected by journalism and by scientific journals. The basis of the study is a full
survey of the almost 1,000 study results that, according to their *Altmetric* Mainstream-Media-
Scores, met with noteworthy resonance in international journalism in the period between
2014 and 2018.

Thus, this study belongs to a group of empirical works that search for robust correlations
between selection decisions in journalism and those in research that can plausibly be inter-
preted as the result of external resonance, namely the popularity of a topic, without, however,
being able to convincingly isolate journalism as the cause of the correlation. The most impor-
tant explanation for this difficulty lies in the 'dual role' of journalism, which changes the world
while describing it [5]. By selecting certain events or statements for news coverage, journalism
increases their (perceived) relevance and social popularity. At the same time, journalism has
reasons for making certain events or statements popular, usually because it believes them to be
true, new and socially relevant [6, 7]—at least in the case of quality journalism. Only events
and statements of this kind promise broad attention and are suited to establishing or strength-
ening the audience's trust in journalism at the same time [8–10]. By choosing socially popular
topics and concurrently making topics socially popular, in systematic studies we cannot in
principle distinguish whether an event or topic that receives journalistic attention also receives
scientific attention because of a publicity effect (through media coverage) or because of a paral-
lel in the attribution of truth, novelty and relevance in journalism and research (earmark
hypothesis). According to the latter, news media "cover certain scientific studies because of
their intrinsic value" [1]. In this reading, news media and scientific journals have similar selec-
tion criteria, namely socially relevant topics. This would at least speak for a *correspondence* in
topic attention between news media and scientific journals. In terms of the publicity effect,
popular media coverage has a *causal* effect on the scientific system, as it provides a scientific
attention boost for scientific studies or topics that they would not receive without news media
coverage [1]. When a scientific study breaks through a certain media reporting threshold, the
perceived importance of the respective topic within the scientific community increases and
thus the likelihood that the topic will be published in scientific journals in the future.

However, proving a causal link would mean assuming that science and journalism apply
fundamentally different standards of evaluation to the question of what is to be considered
true, new and relevant. Apart from numerous anecdotes, there are only a few individual exper-
imental findings that indicate completely different standards for selecting topics [11, 12].
Besides these individual cases, however, it cannot be assumed that journalism and research
apply entirely different standards of evaluation. Aside from theoretical considerations [13],
this assumption can be substantiated by the large number of studies that consistently show
journalism's preference for study results that have been published in respected scientific jour-
nals [14–17].

Accordingly, we must assume that the selection criteria in (quality) journalism and in
research are fairly similar (true, new and relevant events). That said, one cannot completely
separate a publicity effect from the earmark hypothesis, because according to the publicity
effect, scientific journals focus on topics that are socially relevant—which is also the reason
why news media engage with these topics.

Another difficulty in isolating the influence of journalistic selections on internal science
communication is not dissimilar. Journalism is no longer the exclusive gatekeeper for public
dissemination. It has been joined by social media and other online channels, which, depending
on the scientific discipline, are used by scientists, sometimes to a very considerable extent for a
variety of motives, including promoting their own study results [18–20]. Despite intensive
efforts in political communication, it has not yet been possible to isolate journalistic influence
on topic prioritization from that of social media. "Results show that not only do the traditional

media agenda, the social media agenda of parties, and the social media agenda of politicians influence one another but, overall, no agenda leads the others more than it is led by them" [21]. This indicates, all in all, a strong mutual coupling of agendas in journalism and social media [22–24]. Such strong collinearity makes it extremely challenging, if not impossible, to isolate a journalistically mediated public sphere and its impact from that of social media.

Despite these substantial limitations, empirical studies that search for robust correlations between the selection behavior of journalism and that of researchers and research journals are still relevant. They can help to assess the existence, extent and dynamics of a coupling between the public prominence of topics and research and thus contribute to a better understanding of the relationship between external and internal science communication. While not ignoring the substantial limitations described, such a connection could be probably interpreted as the influence of a topic's social popularity—for which popular journalism is still the most valid indicator—on research communication.

In the following sections we put our study into context and describe its methodological design, including data collection, processing and analysis. We then present and discuss the results of our analysis and outline avenues for future research.

## 2 Hypothesis and research question

In the present study, we want to investigate whether there is a relationship between the research topics chosen by international journalism and the topics chosen by scientific journals. Using a time comparative analysis, we examine how a scientific topic develops in scientific journals over time, that is, before and after a point of conspicuous popular attention for a scientific paper on the topic. We try to assess whether the scientific topics that receive popular attention are also in the ascendant in scientific journals and become prominent there. Where we found a significant increase in the attention being paid to a topic in scientific journals after a certain scientific paper had attracted popular attention, this suggests a correspondence between the social popularity of a topic—indicated by its selection by journalism—and that of scientific journals. We hypothesize that there is a correlation between the attention for a scientific research topic in popular media and in scientific journals, as we assume that both are oriented to socially relevant topics:

H1: There is a positive correlation between the external popularity of a research finding—indicated by noteworthy media coverage—and its internal scientific popularity—indicated by a significant change in the number of similar studies published after noteworthy media coverage.

To our knowledge, testing this assumption would be the first attempt to link the external popularity of a topic with the selection behavior of scientific journals.

Moreover, we want to establish whether there is a difference in the degree of correspondence (between popular media and scientific journals) regarding various types of scientific journal. Several studies have already shown that science journalism, when researching and selecting scientific news, is particularly dependent on publications by specific scientific journals and bases its selection decisions predominantly on a very few renowned scientific journals (the relevant studies usually name between seven and ten journals as most frequently cited by journalism). They name, for example, *Nature*, *Science*, *New England Journal of Medicine* (NEJM), *Proceedings of the National Academy of Sciences of the United States of America* (PNAS), *Journal of the American Medical Association* (JAMA), or the *British Medical Journal* (BMJ) [14–17, 25–31]. Two handfuls of scientific journals serve as primary sources for science journalism and may be therefore called 'opinion-leading journals' [32, 33]. These journals

show a proven relationship or correlation to news media coverage and thus seem to be more strongly interwoven with the news media than less often cited scientific journals. We do not know whether the correspondence between popular media coverage and scientific journals is perhaps limited to just these few influential journals. We therefore formulate an open research question differentiating between scientific journals which we would call 'opinion-leading journals' and 'non-opinion-leading':

RQ: Will the degree of correspondence between popular media and scientific journals be higher in opinion-leading journals compared to non-opinion-leaders?

## 3 Method

### 3.1 Data collection and operationalization

To test our hypothesis and research question, we collected different types of data. To investigate the potential short time correspondence between the attention of research topics in popular media coverage and in scientific journals, we apply a systematic approach in which the single scientific article and its (popular) news media coverage are the starting point.

First, we needed to identify scientific research articles that have become popular. We argue that the selection of a study by single individual media titles does not indicate a broader social dissemination of a finding. Such a conclusion seems justified to us only where a single study can gain an appreciable amount of media attention.

\To identify research articles that have become popular and received conspicuous news media attention, we use data from the altmetrics provider *Altmetric*, as other studies have done, e.g. [1] or [34]. More precisely, we use their Mainstream-Media-Score (MSM-Score). *Altmetric* is an increasingly popular online database that reports the number of news outlets, the numbers of tweets, blogs and Facebook pages citing scientific studies. The MSM-Score represents the number of online media portals that mention the respective scientific study. In a previous study, we validated this score to determine the level at which the MSM-score could be interpreted as an indicator of noteworthy popular media coverage of a scientific research article. Our validation was based on 1,601 scientific articles that were published in the journals *Nature* and *Science* between January and October 2017 (extracted from *Scopus* database). We collected the MSM-scores of these studies and assigned them to 11 groups, namely scores of 1–9; 10–19; 20–29;. . .; $\geq$ 100. Subsequently, we conducted a manual search of randomly selected sets of five research articles per group (N = 55) in the full-text press database *Nexis* to determine from which score we could infer journalistic coverage on three large national media markets (the United Kingdom, the United States, and Germany). The amount of press coverage was classified as "noteworthy" if more than 15 articles on at least one of the five studies per category had appeared in these media markets. Our results showed that only a score $\geq$ 50 indicates that single media titles pick up a scientific paper. From a score of $\geq$ 100 it can be assumed that a result has been taken up congruently by a larger number of media titles in different countries (USA, UK, and Germany), indicating a broad international dissemination [28]. We rely on this value $\geq$ 100 as a pure, binary cutoff criterion that is intended to state whether a scientific paper has received noteworthy international media attention or not. We do not carry out any comparative analyzes in which the MSM-score represents an independent variable. We simply used the MSM $\geq$ 100 score to determine our population and identified all scientific papers with an MSM-Score $\geq$ 100 that were published during a time period from August 2014 to July 2018 by applying an automatic search on *Altmetric* using a validated *Python* script [35]. The collection of all scientific papers

that met the cutoff criterion took place in May 2019. These were 1,068 scientific articles from 261 scientific journals.

Second, to operationalize correspondence between popular media coverage and scientific publications, we analyze if the general topic of a scientific article that received a noticeable amount of public attention (operationalized via the MSM-Score $\geq$ 100) also significantly increased within scientific journals in the aftermath (H1).

To capture potential increases of topics within scientific journals, we used the 'similar articles'-function of *PubMed*. *PubMed* is a database that records more than 32 million abstracts and citations of literature from biomedicine, life sciences, chemical sciences, behavioral sciences and bioengineering [36]. *PubMed*'s similar articles-function shows all documents that are the most similar (in terms of content) to the original document you searched for. This is done by a word-weighting algorithm that basically compares words from the title, the abstract, and the so called 'MeSH terms' that are usually added to a document. MeSH is a comprehensive, controlled vocabulary for indexing journal articles and books in the life sciences. For a more detailed description of the process of identifying similar articles see [37]. We validated the precision of the similar articles-function by manually checking all similar articles which *PubMed* has identified for two research articles from our list of 1,068 scientific papers with an MSM-Score $\geq$ 100. In sum, we checked n = 387 similar articles for their content fit to the topic of the respective MSM-Score $\geq$ 100 paper. We found that in one case 46 percent of all analyzed similar articles fit to the topic of the original paper and in the other case 51 percent.

In a third and last step, to answer our research question, we defined which scientific journals we regard as 'opinion-leading journals' (to compare their degree of correspondence to the popular media agenda with those of 'non-opinion-leading journals'). By opinion-leading journals we mean first of all that these scientific journals enjoy a special reputation among science journalists. Secondly, they also enjoy a great reputation in the scientific community. As measurable criteria to determine scientific opinion-leading journals, we 1) chose those journals that have published together at least one third of all MSM-Score $\geq$ 100 papers from our list of 1,068 papers. Further, these journals must 2) noticeably differ from all other journals that published MSM-Score $\geq$ 100 papers, so called 'outliers'. A simple and typical way to identify outliers is to determine them based on the number of standard deviations from the mean. Our criterion to identify outlier-journals was to only choose those journals that differed more than three standard deviations from the mean regarding their number of published MSM-Score $\geq$ 100 papers. The journals that fulfill both of the aforementioned criteria are: *Nature*, *Science*, *New England Journal of Medicine* (NEJM), *Proceedings of the National Academy of Sciences of the United States of America* (PNAS), *The Lancet*, and *Journal of the American Medical Association* (JAMA). These six journals together published 379 of the 1,068 MSM-Score $\geq$ 100 papers between 2014 and 2018 which corresponds to about 35.45 percent.

That this selection of opinion-leading journals also seems to be meaningful in terms of content is the fact that they highly correspond to those journals that are repeatedly named as the scientific journals on which journalists base their selection of scientific papers [14–17].

## 3.2 Data preparation

Of the 1,068 research articles with an MSM score of $\geq$ 100, 983 could be identified in the *PubMed* database. That not all of the 1,068 papers are included in the *PubMed* database is mainly due to the fact that this database is a collection of publications from biomedicine, life sciences, chemical sciences, behavioral sciences and bioengineering. Publications from other disciplines are only covered to a limited extent. At the time of the data retrieval (January 2021), a total of 185,166 articles were assigned to the 983 articles as so called 'similar articles'; since a

variety of articles could be assigned to several of the 983 MSM score $\geq$ 100 articles (because they deal with the same topic), this number is reduced to 157,643 unique papers.

In a next step, we counted the number of similar articles before and after the publication of each associated MSM $\geq$ 100 article. Due to the limitations of the *PubMed* data, only the publication year (contained in the 'PubYear' variable) is available as publication date (no specific indication of month or day). Similar articles that appeared in the same year as the associated MSM $\geq$ 100 article were therefore not considered for analysis. In order to analyze symmetrical observation periods, only those similar articles were considered that were published in the two years before and after the publication of the MSM $\geq$ 100 article. We have chosen the observation period of two years for the following reasons:

Firstly, we had to consider how long it takes till scientists can react to popular media coverage by conducting studies that correspond to scientific topics covered by popular media and to publish them within a scientific journal. Several studies that investigate publication delays—that is delays from submission to publication of an article in a scientific journal—show that they turn out to be very different, depending on the journal itself, the scientific discipline, and several other factors [38]. Further, an increase in the publication delay over time can be observed [39]. For journals like *Nature* and *Science* we find time spans from submission to publication around 100–120 days [39], for biomedical journals of nearly ten months, and for the social sciences even around 24 months. Across various disciplines the publication delay lies between 6–24 months (twelve months on average) [38].

Secondly, most of the studies on publication delays look at the delay on a journal basis, not at an individual paper basis. This means that the publication delay values must usually be obviously higher for individual papers, as they are ordinarily not submitted to one journal only, but to several [38]. Hence, we conclude that we must consider an observation period of a minimum of twelve months (the average value across disciplines according to [38]). As we analyze publications in a broad range of scientific journals, in several cases twelve months will be calculated too short, as the overview of [38] shows. Therefore, we decided to use a longer observation period of two years.

This decision thirdly and additionally follows the idea that lies behind the calculation of Journal Impact Factors that also measure a citation window of two years, which has been empirically validated [40, 41]. Here, too, it is assumed that it will take about two years until a new paper is created that can cite the 'original article'.

Further, we have taken into account that the publications recorded by the *PubMed* database have steadily increased from 532,423 in 2000 to 1,617,971 in 2020. The simple ratio is biased in favor of later time points by this effect if the proportion of publications on the topic remains constant. For this reason, we use a corrected variant in which the number of similar articles published in a year is weighted by the number of publications included in *PubMed* (e.g. $1/n_{2005} + 1/n_{2005} + 1/n_{2006} + 1/n_{2007} + \ldots$). When comparing papers in a particular journal, we used the publication counts of that journal.

## 4 Data analysis and results

To test our hypothesis, we compare the quotient of the number of similar articles related to a certain research topic published by scientific journals during the period before and after the publication of an MSM $\geq$ 100 article (that received noticeable popular media attention) as a simple measure for change.

Assuming H1 that there is a positive correlation between external and internal popularity, we expect an increase in the number of similar articles after an MSM $\geq$ 100 paper was published. To prevent the general increase in the number of published scientific articles from

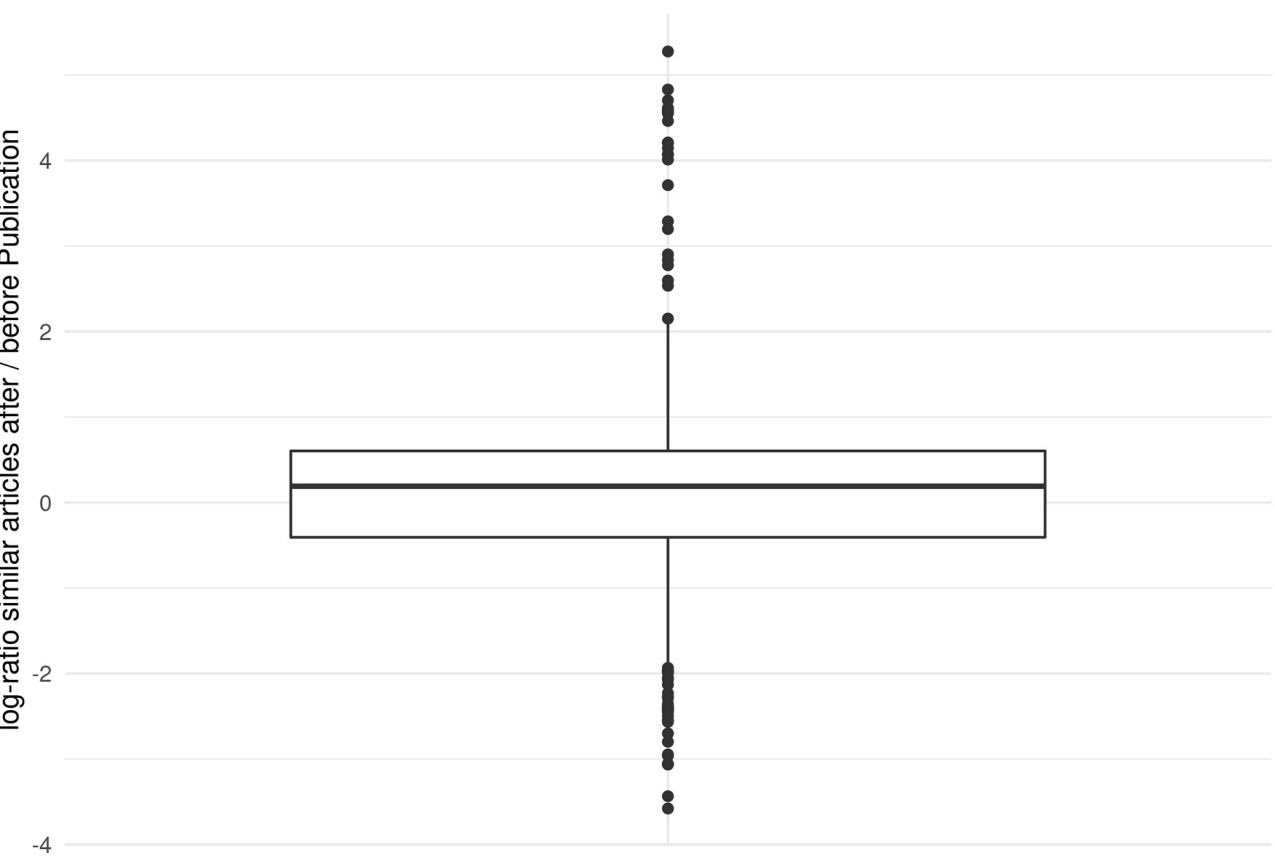

**Fig 1. Increase respectively decrease of similar articles after publication of each MSM ≥ 100 paper.** Based on: 983 MSM ≥ 100 papers. 15 Papers needed pseudo counts.

leading to this effect, we used the values weighted by the annual publication numbers in the analysis.

In the analyses, there are always cases where no similar articles could be found for a paper either in the period before the publication of an MSM ≥ 100 paper or afterwards. In order that a weighted value can also be calculated in this case (it is not mathematically possible to divide n = 0 similar articles by an unknown number of annual publications (because no publication year is available)), pseudo counts are used. The year with the highest publication numbers was chosen as the weight (1/ 1,617,971) so that a weight as small as possible is used, namely 0.000000618058.

Fig 1 shows the box plot of the distribution of the quotients of the number of publications before and after the MSM ≥ 100 paper. That the median line is above zero means that in more than half of the MSM ≥ 100 papers more similar articles on the topic were published in scientific journals after their publication than before. At the median, weighted publication numbers in the thematically related topic area increased by 21 percent in the two subsequent years after an MSM ≥ 100 paper was published. In sum, there was an increase of thematically relevant publications in 59 percent of all cases after popular media coverage. Because of the outliers in the ratios between the number of cases before and after the MSM ≥ 100 paper, we did not test parametrically. A Wilcoxon signed rank test was able to reject the null hypothesis of symmetry around zero (p < 0.01). H1 therefore can be confirmed.

To cite a few concrete examples from our data: The topics of the top ten scientific papers that were published more frequently in scientific journals after popular media attention than before now appeared there 65–199 times more frequently. All of these top ten papers date from 2016 and dealt with the Zika virus, an infectious disease declared a public health emergency of international concern by the *World Health Organization* (WHO) in February 2016. This is an example that shows very well that both journalism and scientific journals select socially relevant topics. Also among the top 20 papers whose topics were covered 18–199 times more frequently in scientific journals than before popular media attention, the Zika virus topic dominates (only two papers are on other topics, the prescription opioid and heroin crisis and polymyxin resistance). Concerning the Zika virus, one can assume that, due to its acute social relevance at the time, it has attracted popular media as well as scientific attention. The popular media picked up the Zika topic relatively early (at a time when only a few scientific Zika studies had been published). An increased scientific attention followed later. Other scientific topics that received increased attention in popular media and in the scientific community afterwards were colorectal cancer (13 times more frequently published after popular media attention), migraine (12 times more frequently published), cometary science (11 times more frequently) or atopic dermatitis (8 times more frequently), to name just a few examples.

The general effects observed by our data are rather small at the aggregated level. On median, eleven similar articles were published for one MSM $\geq$ 100 paper both before and after its publication. An increase of 21 percent would therefore mean that only two more publications on the respective topic were published after the MSM $\geq$ 100 paper (overall average). In individual cases this increase can of course be considerably higher.

Of course, it should also be mentioned that there were scientific papers as well whose topics were published less frequently in scientific journals after popular media attention than before. In these cases, we cannot speak of a correlation between the choice of topics in mass media and in research—although the topics of the scientific papers do not seem to be socially irrelevant. Among the ten papers that most often had fewer similar articles published after popular media attention (11–33 times fewer) than before, are the topics: Zika virus, influenca vaccine, clean energy, breast cancer, noninvasive blood tests for fetal development and others. All of these ten papers have been published between 2017–2018, when the Zika virus, for example, was no longer so acute. In these cases, the popular media have paid attention to the respective research topics (much) later than scientific research (a certain body of scientific publications already existed at the time of the media coverage).

In a next step, we want to answer our research question and differentiate between scientific opinion-leading journals and non-opinion-leading journals. We ask, if the confirmed correspondence between the attention for a scientific research topic in popular media and in scientific journals occurs in scientific opinion-leading journals in the same way as in non-opinion-leaders. To test our research question, we make two comparisons: First, we test whether opinion-leading journals publish more thematically similar articles after a topic received popular attention compared to non-opinion-leading journals. If this were the case, then one could say that the correspondence between journalism and opinion-leading journals is higher than between journalism and non-opinion leading journals. Second, we check if it makes a difference whether the paper that received popular media attention (the MSM $\geq$ 100 paper) was originally published in an opinion-leading journal or in a non-opinion-leading journal. If MSM $\geq$ 100 papers originally published in opinion-leading journals triggered a greater increase in the publication of thematically similar articles in scientific journals than MSM $\geq$ 100 papers originally published in non-opinion-leaders, then it would appear that characteristics of the publishing journals (e.g. their scientific standing) play a role in the attention given to a topic by the scientific community.

The first-mentioned comparison between opinion-leading journals and non-opinion-leaders regarding their quotients of similar articles before and after the publication of MSM $\geq$ 100 papers is not that easy. At the median, weighted publication numbers in the thematically related topic area increased in opinion-leading journals by 50 percent in the two subsequent years after an MSM $\geq$ 100 paper was published, in non-opinion-leading journals the increase was 30 percent. A Wilcoxon rank sum test for location shift between the two distributions was not significant (p = 0.3). In particular, for the opinion-leading journals the number of similar articles was very small, so the analysis is problematic in some respects. Only in 317 (of 983) cases similar articles were found in opinion-leading journals before and after the publication of a MSM $\geq$ 100 paper. For 309 publications no similar articles were found in opinion-leading journals before *or* after the publication of MSM $\geq$ 100 papers. In 357 cases no similar articles were found in opinion-leading journals before *and* after the publication of MSM $\geq$ 100 papers. This leads to the fact that the method for the calculation of pseudo counts has a direct influence on the first and third quartile, which are essential for the calculation of box plots. Therefore, we cannot clearly answer our research question. We can only state that we see no evidence in our data that opinion-leading and non-opinion-leading journals behave differently.

With regard to the second-mentioned comparison, we checked whether the place of publication of a scientific paper that received popular attention plays a role. Do scientific journals pay more attention to publicity papers (MSM $\geq$ 100 papers) if they were originally published in an opinion-leading journal? The idea behind this test is that we already know from previous analyses that especially scientific opinion-leading journals, who enjoy a high reputation among scientific actors, steer the attention of science internally, so that probably other scientific journals also orient (e.g. their topic selection) towards them (e.g. Franzen, 2011). So, if we would *not* find an increase of thematically related publications in scientific journals after the publication of MSM $\geq$ 100 papers from non-opinion-leading journals, then we could not assume a general or robust correspondence between the choice of topics by mass media and research. Rather this would speak for an effect of the reputation of the journal that published the paper that generated popular media attention. For our analysis, we separated the MSM $\geq$ 100 papers examined into the 380 papers that appeared in one of our six defined opinion-leading journals and the 603 papers that did not and run the same analysis as before.

Fig 2 shows the bean plots of the two distributions. We see that the median number of papers published on a certain topic increased by 28 percent for the MSM $\geq$ 100 papers from non-opinion-leading journals, while the median increase for the other papers was 39 percent. A Wilcoxon rank sum test for location shift between the two distributions was not significant (p = 0.1). Both scenarios lead to an increase in the proportion of thematically related publications within scientific journals (median line above zero in both cases). Thus, the data do not speak against hypothesis 1.

In a last step, as a control group design, we test whether there also is a considerable increase in thematically relevant papers after a scientific paper has *not* received any noteworthy popular media attention (but has 'only' appeared in a scientific opinion-leading journal). Such a finding would suggest that not social popularity of the topic but the reputation of the publishing journal triggers the increase of thematically related publications. This would speak against a correspondence between the choice of topics in popular media and scientific journals. For the analysis, we have created a dataset that consists of all scientific papers that have been published in the two most prestigious [42] scientific opinion-leading journals *Nature* and *Science* in the same time period (2014 to 2018) with an MSM-score less than 50. These papers should according to [28] not have received a noticeable amount of popular media attention. The idea behind the limitation of our analysis to *Nature* and *Science* is the following: If we were able to find a journal effect, it would be in these two journals, since they represent the most prestigious

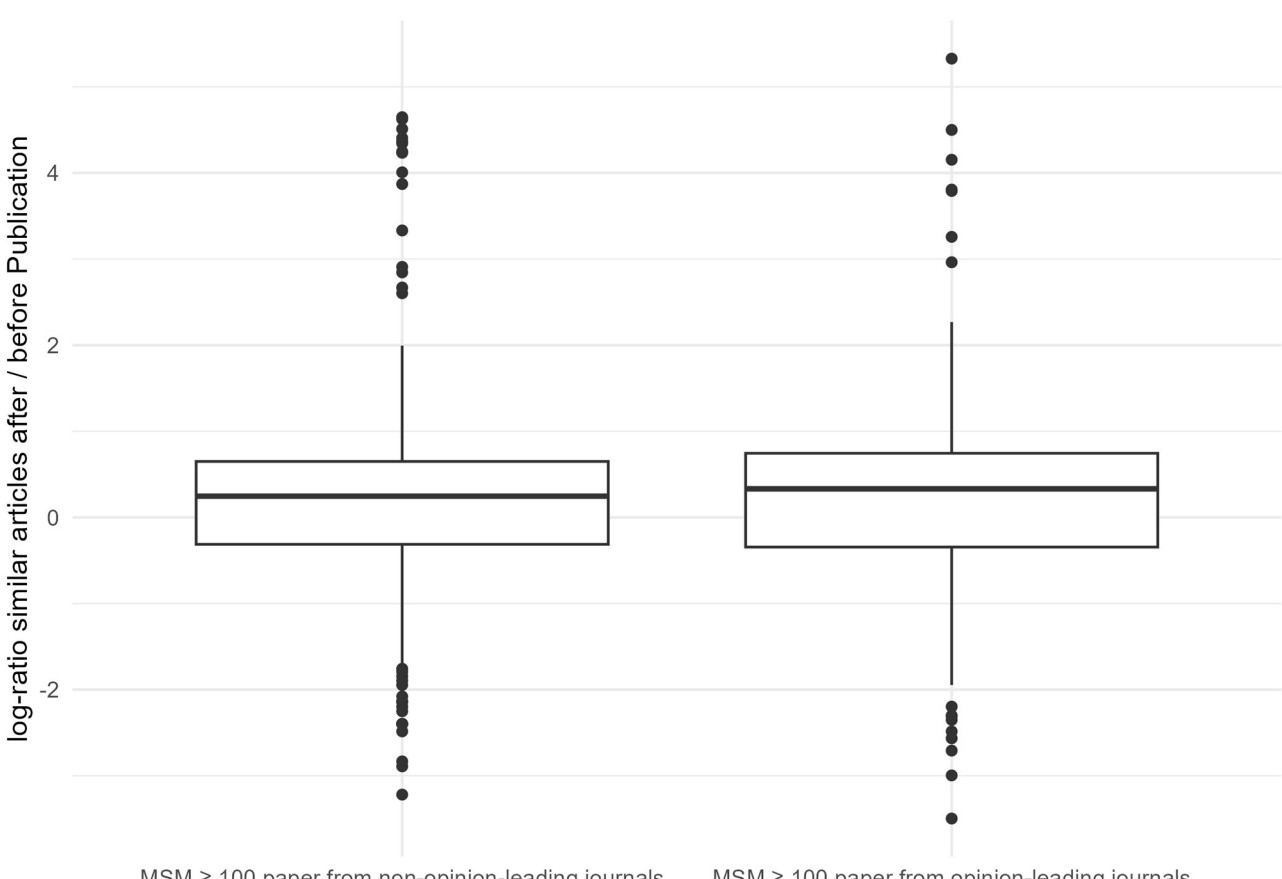

**Fig 2. Comparison of increase of similar articles after publication of each MSM ≥ 100 paper originating from non-opinion-leading or opinion-leading journals.** Based on: 603 MSM ≥ 100 papers from non-opinion-leading journals and 380 from opinion-leading journals.

scientific journals. Adding other but less prestigious journals would rather water down the results. We run the same analyses for these 4,824 papers, as before and thus compared the quotient of the number of similar articles related to the research topics from the period before and after the publication of an MSM < 50 article published by *Nature* or *Science*. We then compared the increase of similar articles of the MSM < 50 articles from *Nature* and *Science* with the increase of similar articles of the MSM ≥ 100 articles from *Nature* and *Science*.

Fig 3 reveals a significant difference in the two distributions. While the MSM ≥ 100 articles showed a median weighted increase of similar articles of 12 percent, the papers with MSM scores < 50 showed a decrease of 15 percent (median line below zero). A Wilcoxon rank sum test showed significant differences (p < 0.01). This finding shows that there is a difference between those papers that received a noticeable amount of public media attention and those which did not. For us, this is another clear indicator that our first hypothesis is confirmed, namely that the topic selection of popular media and scientific journals are correlated. Scientific topics that attract public attention also attract more attention in scientific journals—probably because they are socially relevant.

## 5 Summary and discussion

The aim of our analysis was to expand the body of empirical research on topic selection in scientific journals and to investigate whether there is a robust correlation between journalism's

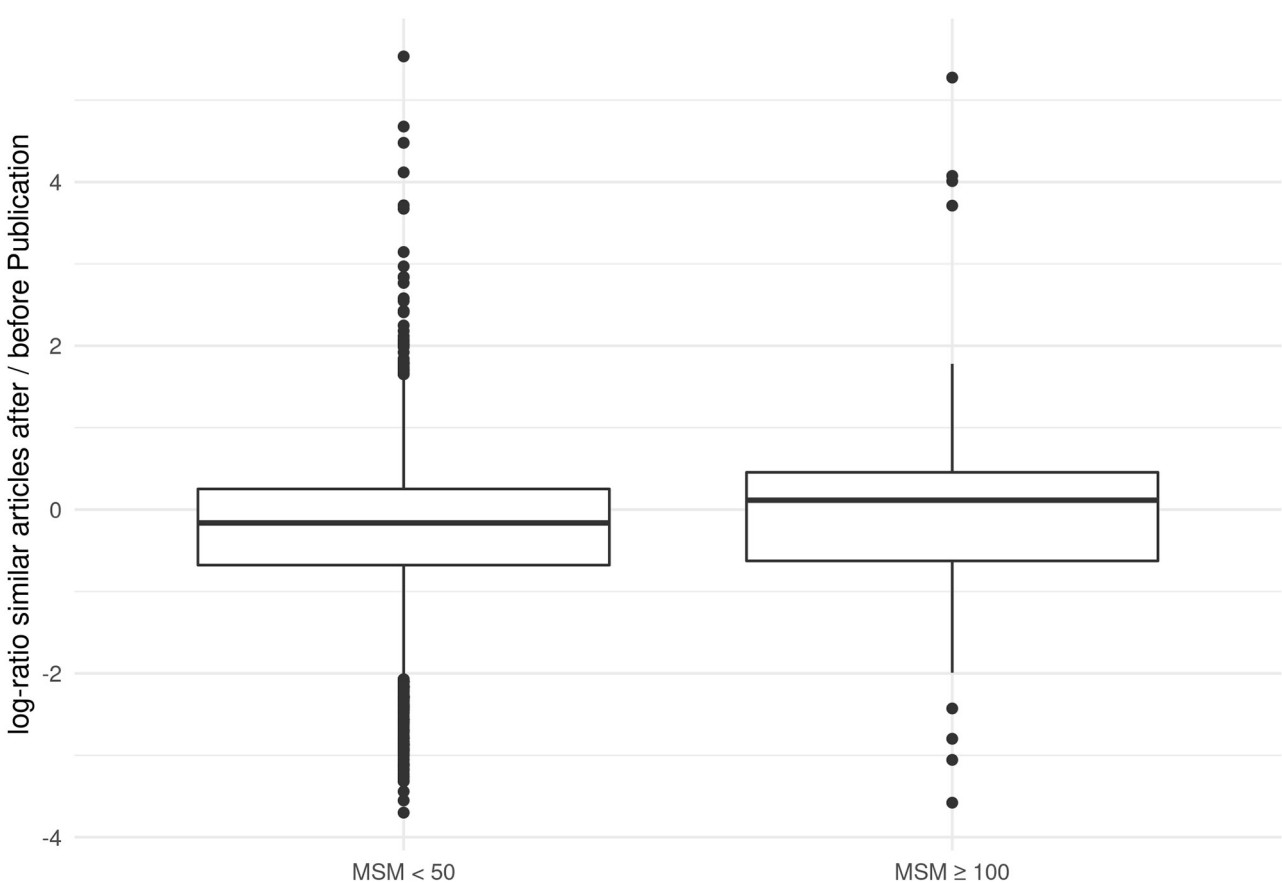

**Fig 3. Comparison of increase respectively decrease of similar articles after publication of MSM < 50 respectively MSM ≥ 100 papers from** *Nature* **or** *Science.* Based on: 179 MSM ≥ 100 papers and 4,824 MSM < 50 papers from *Nature* and *Science.*

topic selection and that of scientific journals, making it plausible that the popularity of a topic potentially influences the research agenda. In sum, our results indicate a robust correlation between the choice of topics in mass media and in research. At the aggregated level, our analysis showed that, after a scientific paper that received a noticeable amount of popular media attention (MSM ≥ 100 paper) was published, in more than 50 percent of the cases more similar articles on the topic were also published in scientific journals. Our hypothesis, that is, that there is a positive correlation between the external popularity of a research finding—indicated by noteworthy media coverage—and its internal scientific popularity—indicated by a significant change in the number of similar studies published after noteworthy media coverage—is thus confirmed. In this context, we found no significant differences between scientific opinion-leading journals and non-opinion-leaders—for both types of journal we identified a correlation between popular media attention and scientific journal publishing. Furthermore, the correlations between popular media attention and scientific journal publishing seem to be relatively robust: they apply to both scientific papers that were originally published in renowned opinion-leading journals as well as in less renowned non-opinion-leaders but not to scientific papers that had not received a noticeable amount of news media attention. The latter finding, in our logic, indicates that socially unpopular topics attract neither great journalistic nor scientific attention.

We believe that the results of our study constitute evidence for the earmark hypothesis, namely that the correlation between the choice of topics in mass media and in scientific journals is due to the parallelism of relevance attributions and selection criteria in journalism and research. However, we cannot completely separate a publicity effect from the earmark hypothesis, because according to the publicity effect, scientific journals focus on topics that are publicly relevant—which is also the reason why news media engage with these topics. Ultimately, the distinction between the earmark hypothesis and the publicity effect is of secondary importance in our view. What matters is that there is a relationship between the choice of topics in journalism and in scientific journals. Topics of social relevance are probably (more) likely to be picked up by popular media as well as by scientific journals.

When interpreting the results, one should keep in mind that we are talking about a very small proportion of scientific papers which receive notable popular media coverage, that is, approximately one to two papers in 10,000 [28]. Nevertheless, we see a clear and stable pattern in our data that indicates that there is a certain degree of correlation between popular media and journal selection of scientific papers. In order to prove causal feedback effects of the social popularity of a topic on the research or scientific journal agenda, experimental research designs would be appropriate.

While communication science already provides numerous explanations for and studies on journalists' selection processes (e.g. news value studies), we think that there is a lack of empirical work that analyzes and describes the selection behavior of scientific journals (editors), like the work of [43], for example. We believe it would be particularly interesting to observe or survey the scientific journal editors (albeit probably difficult to achieve). As [1] put it: "Much remains to be discovered concerning interactions between popular media, generally produced by and for non-scientists, and the scientific literature which has historically been written by and for the scientific community".

One limitation of our study needs to be mentioned: our validation of the *PubMed's* 'similar articles'-function to capture potential increases in topics in scientific journals showed that the topic fit of this function is not as satisfactory as we would have expected. Consequently, there is a considerable amount of noise in our data, that is additionally increased by the rather unspecific information provided on the publication date of the analyzed papers (no specific indication of month or day in *PubMed*).

Another uncertainty concerns the threshold of the MSM-score, which we defined as an indicator of noteworthy popular media coverage of a scientific research article. We defined this threshold on the basis of a validation study we conducted. However, we cannot say with absolute certainty that the MSM $\geq$ 100 papers do not include papers that have not received noteworthy international media coverage. Ultimately, we cannot provide a very specific measure of when a scientific study actually received broad international media attention.

Regarding the MSM $\geq$100 articles, we should also state that, in our sample, they are not evenly distributed over the analyzed publication years. In 2014 and 2015, for example, we have significantly fewer MSM $\geq$100 articles in the sample than in the years 2016–2018. Thus, there is time bias in our data. We suspect that this is because, over the years, *Altmetric* has changed respectively expanded the source base they track to capture references to scientific papers. However, as we have not carried out any stratified analyzes broken down by year of publication, we do not consider this to be a problem for our analyses. Moreover, we have checked whether the different case numbers in the individual years have an impact on our results. An analysis using only the MSM $\geq$100 articles from the years 2016–2018 (excluding the years with the low number of cases) resulted in no change in our key findings. Our findings thus remain relatively stable, which is plausible as only relatively few cases were excluded from the analysis.

## Acknowledgments

We thank Nikolai Promies, Clarissa Staudt and Marie Thederan for assistance with sample creation and data collection.

## Author Contributions

**Conceptualization:** Melanie Leidecker-Sandmann, Lars Koppers, Markus Lehmkuhl.

**Data curation:** Melanie Leidecker-Sandmann, Lars Koppers.

**Formal analysis:** Lars Koppers.

**Funding acquisition:** Markus Lehmkuhl.

**Investigation:** Melanie Leidecker-Sandmann, Lars Koppers, Markus Lehmkuhl.

**Methodology:** Melanie Leidecker-Sandmann, Lars Koppers, Markus Lehmkuhl.

**Project administration:** Melanie Leidecker-Sandmann, Markus Lehmkuhl.

**Resources:** Lars Koppers.

**Supervision:** Markus Lehmkuhl.

**Visualization:** Lars Koppers.

**Writing – original draft:** Melanie Leidecker-Sandmann, Lars Koppers.

**Writing – review & editing:** Melanie Leidecker-Sandmann, Markus Lehmkuhl.

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
