## [Decision Letter · Decision Letter 0]

4 Aug 2022

PONE-D-22-17799Attention boost through media coverage? Agenda setting effects from news media coverage on topic selection of scientific journalsPLOS ONE

Dear Dr. Leidecker-Sandmann,

Thank you for submitting your manuscript to PLOS ONE. After careful consideration, we feel that it has merit but does not fully meet PLOS ONE’s publication criteria as it currently stands. Therefore, we invite you to submit a revised version of the manuscript that addresses the points raised during the review process.

The reviewers found this to be an interesting study, exploring a novel research question. However, they each raised some methodological critiques, which I ask you to address to ensure the validity and rigour of your conclusions. Particularly, there are concerns that the conclusions, methods, and research question are not congruent, undermining the potentially interesting and important findings. Please address each of the reviewer's comments, but specifically the questions related to the Altmetric and MSM measurements to ensure the validity of your measures of attention; and please consider the use of comparative statistical methods to support your results and revise your conclusions accordingly. I look forward to receiving your revised manuscript.

We look forward to receiving your revised manuscript.

Kind regards,

Quinn Grundy, PhD, RN

Academic Editor

PLOS ONE

Journal Requirements:

"This work was supported by the German Research Foundation (DFG), Bonn within the framework program “Medializing brain diseases: interactions between research and mass media” (Reference No. 411038189) https://gepris.dfg.de/gepris/projekt/411038189?language=en and by the Volkswagen Foundation in the framework program “Entwicklung von Methoden und Tools für eine datengestützte Wissenschaftskommunikation” [“Development of methods and tools for data-based science communication”, Hannover (Reference No. 94838) https://www.volkswagenstiftung.de/. The author who received the award was M. L.

3. Please amend your manuscript to include your abstract after the title page.

4. We note that you have referenced (Dumas et al. [9]) which has currently not yet been accepted for publication. Please remove this from your References and amend this to state in the body of your manuscript: (“Dumas et al. [Unpublished]”) as detailed online in our guide for authors

Reviewers' comments:

Reviewer's Responses to Questions

**Comments to the Author**

1. Is the manuscript technically sound, and do the data support the conclusions?

Reviewer #1: No

Reviewer #2: Partly

2. Has the statistical analysis been performed appropriately and rigorously? 

Reviewer #1: No

Reviewer #2: No

3. Have the authors made all data underlying the findings in their manuscript fully available?

Reviewer #1: Yes

Reviewer #2: Yes

4. Is the manuscript presented in an intelligible fashion and written in standard English?

Reviewer #1: Yes

Reviewer #2: Yes

5. Review Comments to the Author

Reviewer #1: This study compares the scientific productivity related to specific research topics before and after the release of a considerable number of news coverage about those topics. Regarding the research questions and the methods used, I have the following questions:

1. The ambition of the authors is to examine the feedback effect of news coverage on scientific productivity. However, there must be a variety of factors that play a role in affecting the productivity of scientific papers around a specific topic. In the case of altmetrics, for example, in addition to news media, other channels such as blogs, Twitter and Facebook can also increase the visibility of scientific papers as well as their related topics. More importantly, these data sources show a similar pace of data accumulation as news media – most of the data from these sources are accumulated in the early stage after the publication of papers. In other words, although the authors used news coverage counts to divide “similar articles” into two periods (i.e., before/after), similar results may have been observed with other altmetric data sources as the objects of study (e.g., blogs and Twitter). Therefore, comparative analyses with other altmetric data sources as the research objects are needed, to rule out the influence of other altmetric factors and to validate the conclusion that news coverage plays a key role in influencing the thematic trends of scientific journals.

2. Although the authors mentioned that their findings were unable to “draw the causal conclusion”, the hypothesis – “a scientific topic that attracted media attention will attract more attention in scientific journals” – sounds more like an assumption of a causality between news media attention and subsequent scientific productivity, particularly when the authors concluded that “journalism can be considered as an agenda setter for the choice of topics in academic journals”. If the authors aimed to verify this hypothesis, specific methods of causal inference should have been employed, rather than merely the observations based on box plots.

3. “Since the agenda setting approach is considered to be relatively well scientifically proven and since correlations between popular media coverage and citation rates of scientific studies have already been evidenced” This argument seems to be one-sided because there have also been many studies based on larger dataset reporting weak correlations between news media counts and citations at the scientific paper level, for example,

DOI: 10.1002/asi.23309

DOI: 10.1371/journal.pone.0120495

4. The “similar articles” function of PubMed was adopted to measure the scientific productivity around specific topics. However, only nearly half of the similar papers identified by PubMed were correct as the authors manually checked. Is this precision degree sufficient to support the idea that this function can be used to represent the scientific productivity related to certain research topics? Especially given that the authors also mentioned “here are always cases where no similar articles could be found”. I am wondering how much of an impact these flaws would have.

5. As for Fig 2 and Fig 3, a significance test of the difference between the two groups is needed to support the arguments of “small difference” and “significant difference”.

6. For Fig 3 specifically, I am wondering why not just continue to use the six journals as the so-called opinion-leader journals and instead selected a new dataset that included only Nature and Science.

7. Please pay attention to the typos, like “Altmetric” in P9, L205.

Reviewer #2: Although the research questions proposed by the authors are interesting and worth comprehensive study, there are some problems with this manuscript as follows:

i) It takes some time for the MSM-score to accumulate to a high number. The MSM-score of the paper will not necessarily reach more than 100 points at the time of publication that year, especially for the paper published in December. In order to truly select a high MSM-score article (i.e. an article that received noticeable media attention), the time effect of cross year must be eliminated, and then “compare the quotient of the number of thematically similar articles published by scientific journals from a two-year period before respectively after the publication of an article that received noticeable media attention.” It is suggested to redesign the data collection and processing method.

ii) The analysis method is too simple. Please use the standard statistical method for comparative analysis. Since there is no control group, the conclusions are doubtful. Although there seems to be a little comparison in Fig 3, it is far from enough.

iii) As for the threshold of MSM-score, why choose more than or equal to 100? Can you choose 50 or other standards? This manuscript refers to the Kohler et al.’s paper (2020) which is a preprint without peer review, and one of the authors is lehmkuhl, who is also the author of the manuscript. Therefore, the reason for choosing 100 must be explained in more detail. It is suggested to choose different thresholds, such as 100, 50, 25, etc., and do a variety of studies from different perspectives.

In short, due to some problems in data and analysis methods, the current conclusion of manuscript is questionable.

References

Kohler S, Promies N, Lehmkuhl M. Patterns in the journalistic selection of neuroscientific research results. SocArXiv. 2020. doi: 10.31235/osf.io/s9dy7.

6. PLOS authors have the option to publish the peer review history of their article (what does this mean?). If published, this will include your full peer review and any attached files.

Reviewer #1: No

Reviewer #2: No

---

## [Author Response · Author response to Decision Letter 0]

13 Oct 2022

Dear editors, dear anonymous reviewers!

We would like to thank all of you for the helpful feedback concerning the article we submitted to PLOS ONE. We have taken the editor's and reviewers' comments into account and hope that our changes and explanations are understandable and satisfactory. While revising our article, we felt that it gained in quality and we hope that you feel the same way when reading this letter and the revised article.

Below (Response to Reviewers), we list the editor's and reviewers’ comments sorted by topic and address each point.

(1) Causality issue, accompanied by methodological problems

Reviewer 1, comment 2: Reviewer 1 notes that our theoretical argumentation and hypotheses suggest causal inferences that we cannot prove with the analysis methods we used.

Reviewer 2, comment 2: Reviewer 2 also states that our analysis methods are too simple to test our assumptions. Comparative analyses/ control groups are required.

Reviewer 1, comment 5; Reviewer 2, comment 2: Both reviewers demand significance tests.

Reviewer 1, comment 1: Reviewer 1 further recommends the inclusion of social media data sources to be able to compare if they also have an influence on the topic selection of scientific journals.

Action taken:

We have taken these criticisms very seriously, because they are of a substantial nature for our paper. We agree with the reviewers’ critique. We decided that the agenda setting approach, which assumes causal effects of the media agenda on the scientific agenda, leads in the wrong direction. Therefore, we reframed our theoretical argumentation. We substantially revised chapters 1 (Introduction) and 5 (Summary and discussion), deleted chapter 2 (Feedback effects of media coverage on scientific publishing) and have reformulated hypothesis 1. We now explain that we consider journalistic coverage as an indicator of the social popularity of a topic and focus on the question whether socially popular topics (that have attracted public attention through media coverage) are considered worthier of publication by scientific journals than those which did not gain public attention. Primarily, we seek to plausibilize the existence of a robust connection between topic selection in journalism and research. In our new introduction, we try to make a plausible case that the selection criteria of quality journalism and scientific journals should be similar (earmark hypothesis), namely that both should be oriented toward socially relevant topics. And if this is the case, then one cannot separate a publicity effect from the earmark hypothesis without doubt, because according to the publicity effect, scientific journals focus on topics that are publicly relevant – which is also the reason why news media dedicate themselves to these topics. Ultimately, the distinction between the earmark hypothesis and the publicity effect is of secondary importance in our view. What matters is that there is a relationship between the choice of topics in journalism and in scientific journals.

To make the new theoretical focus of our paper clearer, we have also changed the title of our manuscript to “Correlations between the topic selection of news media and scientific journals”.

Based on our methodological approach, as the reviewers correctly note, causal interpretations cannot be deduced without doubt. Hence, we focus our analysis and interpretation to correlations between the selectivity of journalism and that of research journals. We enriched our analyses with significance tests. Because of the skewed distribution of ratios with strong outliers, we opted for a nonparametric approach and thus fortwo-sample Wilcoxon tests. The analysis of Figure 3 serves as a control group design and is now explicitly named as such. 

The inclusion of social media data sources to test whether they have an influence on the topic selection of scientific journals is no longer necessary due to the new theoretical frame. However, in order to take up this point in substance, we discussed in the manuscript (chapter 1) the strong mutual coupling of agendas in journalism and social media, which makes it extremely challenging if not impossible to isolate a journalistically mediated public sphere and its impact from that of social media. This serves as a further argument why we refrain from a causal view.

(2) Effects of media coverage on citation rates

Reviewer 1, comment 3: Reviewer 1 does not entirely agree with our argument about effects of media coverage on citation rates.

Explanations: 

Due to our new theoretical framework, the objection in comment 3 by reviewer 1 becomes obsolete, since we no longer make the criticized argument. We do not talk about effects of media coverage (on citation rates) anymore.

(3) Precision degree of the “similar articles” function of PubMed

Reviewer 1, comment 4: Reviewer 1 wonders, how much of an impact the rather weak precision of the “similar articles” function of PubMed has on our results.

Explanations: We consider the quotient of the number of similar papers before and after a point in time. A low precision goes into the numerator as well as into the denominator, so the result remains expectation-true, albeit with a higher variance. As long as the precision of the “similar articles” function remains stable over time, it has no influence on the result of our analysis. There is no plausible reason to assume that precision should change over time. 

(4) Threshold of MSM-score

Reviewer 2, comment 3: Reviewer 2 expects a more detailed justification of why we chose the threshold more than or equal to 100 for MSM-scores. Furthermore he/she suggests to select and test different thresholds.

Explanations:

In a previous study, we validated the MSM-score score to determine the level at which the MSM-score could be interpreted as an indicator of noteworthy popular media coverage of a scientific research article. Our validation was based on 1,601 scientific articles that were published in the journals Nature and Science between January and October 2017 (extracted from Scopus database). We collected the MSM-scores of these studies and assigned them to 11 groups, namely scores of 1–9; 10–19; 20–29;…; ≥ 100. Subsequently, we conducted a manual search of randomly selected sets of five research articles per group (N = 55) in the full-text press database Nexis to determine from which score we could infer journalistic coverage on three large national media markets (the United Kingdom, the United States, and Germany). The amount of press coverage was classified as ‘noteworthy’ if at least 19 articles on at least one of the five studies per category had appeared in these media markets. A previous study validated this score and Our results showed that only a score ≥ 50 indicates that single media titles pick up a scientific paper. From a score of ≥ 100 can it be assumed that a result has been taken up congruently by a larger number of media titles in different countries (USA, UK, and Germany), indicating a broad international dissemination [28]. We rely on this value ≥ 100 as a cutoff value to determine our population […].

Action taken:

We have integrated this text passage into our manuscript (chapter 4.1). 

In addition, we have pointed out the following limitation of our study in chapter 5: “Another uncertainty concerns the threshold of the MSM-score, which we defined as an indicator of noteworthy popular media coverage of a scientific research article. We defined this threshold on the basis of a validation study conducted by us. However, we cannot say with absolute certainty that the MSM ≥ 100 papers do not include papers that have not received noteworthy international media coverage. Ultimately, we cannot provide a very specific measure of when a scientific study has actually received broad international media attention.”

Additional explanations: 

We do not use the MSM-scores to conduct any statistical analyses, but only as a cutoff variable that indicates, which research article received popular media attention. We treat the MSM-score as a categorical variable: Either a study has received relevant media attention, or it has not. Therefore, we see no need to test several values.

(5) Accumulation of the MSM-score

Reviewer 2, comment 1: Reviewer 2 thinks that the time it takes for the MSM-score to develop is problematic for our data collection and processing method.

Explanations:

Reviewer 2 considers that „[i]t takes some time for the MSM-score to accumulate to a high number.” Therefore, our data collection and processing method seems problematic for him/her.

We would like to make two counterarguments:

1) We do not share the assumption that it takes about (more than) a year until an MSM-score reaches 100 points. We found several literature references which state that “Altmetric scores react immediately after publication of a study.” [1] See also [2]: “[research] articles may be mentioned predominately during the time period that they are first published.” Also, reviewer 1 argues, that MSM-scores “are accumulated in the early stage after the publication of papers.” The reviewers contradict each other on this point.

Also, from a communication science perspective, it makes entirely sense that news media pick up a scientific study when it is freshly published. When it comes to the question of what is newsworthy, news media and journalists are guided by so-called news factors, among other things (see news value studies, e.g. [3]). One crucial news factor with a strong impact on the selection process of science news is “actuality” [4]. Only rarely are studies that were published months or years earlier selected for science news coverage – for example, if they can be thematically re-updated.

Literature:

1. Studenic P, Ospelt C. Do you tweet? Trailing the connection between Altmetric and research impact! RMD Open. 2020;6(3): e001034. doi: 10.1136/rmdopen-2019-001034

2. Elmore, SA. The Altmetric attention score: What does it mean and why should I care? Toxicologic Pathology. 2018;46(3): 252-255. doi: 10.1177/0192623318758294

3. Maier, M, Stengel, K, Marschall, J. Nachrichtenwerttheorie [News value theory]. Konzepte. Ansätze der Medien- und Kommunikationswissenschaft. Baden-Baden: Nomos; 2010. doi: 10.5771/9783845260365

4. Badenschier F, Wormer H. Issue selection in science journalism. Towards a special theory of news values for science news. In: Rödder S, Franzen M, Weingart P, editors. The Sciences’ Media Connection: Public Communication and its Repercussions. Dordrecht: Springer; 2012. pp. 59–85.

2) Irrespective of whether or not one may follow our argumentation under 1), from our point of view there is no problem with the data collection for the following reason: We retrieved the MSM ≥100 paper all at a consistent point of time, namely in May 2019. Since the publication date of our selected papers had to be between August 2014 and July 2018 (pick-up criterion), the collection date is far enough behind the publication date of the articles, so that an MSM-score should already have developed. The MSM-score ≥ 100 is a cutoff value for us to determine the population: We have only collected papers that already have generated an MSM-score ≥ 100. We are not further interested in the development of the MSM-scores of the selected papers and do not make any analyses with the MSM-score. It solely serves as a criterion for the selection of our population.

Action taken:

We now clarify the use of the MSM-score as a pure cutoff value to determine the population in our manuscript by using the following wording: “We rely on this value ≥ 100 as a cutoff value to determine our population and thus identified all scientific papers with an MSM-Score ≥ 100 that were published during a time period from August 2014 to July 2018 by applying an automatic search on Altmetric using a validated Python script [37]. The collection of all scientific papers that met the cutoff criterion took place in May 2019. These were 1,068 scientific articles from 261 scientific journals.”

(6) Dataset figure 3

Reviewer 1, comment 6: Reviewer 1 wonders why our dataset related to figure 3 consists of only two instead of six opinion-leader journals.

Explanation:

The analysis which relates to figure 3 serves as a control group design to test for effects of highly prestigious scientific journals: We ask if scientific papers that were originally published in highly renowned scientific journals but have not gained popular media attention imply an increase of similar articles in other scientific journals? The idea behind the limitation of our analysis to the journals Nature and Science is the following: If we were able to find a journal reputation effect, it would be in these two journals, since they represent the most prestigious scientific journals. Adding other but less prestigious journals would rather water down the results.

Action taken:

We have included this explanation into our manuscript: “The idea behind the limitation of our analysis to Nature and Science is the following: If we were able to find a journal effect, it would be in these two journals, since they represent the most prestigious scientific journals. Adding other but less prestigious journals would rather water down the results.”

(7) Typos

Reviewer 1, comment 7: Reviewer 1 complains about typos.

Action taken:

We have thoroughly proofread the manuscript and hope to have eliminated all typos.

(8) Additional requirements, correct implementation of PLOS ONE’s style guidelines

Action taken:

We have implemented all noted journal requirements and have reviewed the manuscript with regard to the application of the journal's guidelines.

---

## [Decision Letter · Decision Letter 1]

9 Nov 2022

PONE-D-22-17799R1Correlations between the topic selection of news media and scientific journalsPLOS ONE

Dear Dr. Leidecker-Sandmann,

Thank you for submitting your manuscript to PLOS ONE. After careful consideration, we feel that it has merit but does not fully meet PLOS ONE’s publication criteria as it currently stands. Therefore, we invite you to submit a revised version of the manuscript that addresses the points raised during the review process.

Thank you for your thoughtful and comprehensive attention to the reviewers' comments. Before I can proceed to a final decision, I kindly ask you to address the following minor comments:- the reviewer raises the issue of bias related to time of publication; please address and/or respond to this concern- the abstract well emphasizes the key findings and limitations of the study; you might consider adding further detail related to the sample (including the n), population and methods of study, key statistics to support the key findings (if they can stand alone), and a conclusion that sums up the key relevance- I would recommend having a copy editor work through the manuscript for conciseness and clarity in sentence structure including:  "plausibilizing" and "plausibilize" replaced with "making plausible", breaking up the introduction into smaller paragraphs to aid readability; check missing punctuation (periods) in places following tracked changes removal- The Introduction makes a strong argument for the study, but could be reworked to deliver the key points more succinctly and to remove repetition. A few sentences were unclear to me; could you rephrase these? For example, I didn't understand the point "It is for instance obvious that mass media is important for a change in values regarding animal trials."==============================

We look forward to receiving your revised manuscript.

Kind regards,

Quinn Grundy, PhD, RN

Academic Editor

PLOS ONE

Journal Requirements:

Reviewers' comments:

Reviewer's Responses to Questions

**Comments to the Author**

1. If the authors have adequately addressed your comments raised in a previous round of review and you feel that this manuscript is now acceptable for publication, you may indicate that here to bypass the “Comments to the Author” section, enter your conflict of interest statement in the “Confidential to Editor” section, and submit your "Accept" recommendation.

Reviewer #2: (No Response)

2. Is the manuscript technically sound, and do the data support the conclusions?

Reviewer #2: Partly

3. Has the statistical analysis been performed appropriately and rigorously? 

Reviewer #2: Yes

4. Have the authors made all data underlying the findings in their manuscript fully available?

Reviewer #2: Yes

5. Is the manuscript presented in an intelligible fashion and written in standard English?

Reviewer #2: Yes

6. Review Comments to the Author

Reviewer #2: The authors said that they “retrieved the MSM ≥100 paper all at a consistent point of time, namely in May 2019. Since the publication date of our selected papers had to be between August 2014 and July 2018 (pick-up criterion), the collection date is far enough behind the publication date of the articles, so that an MSM-score should already have developed.”

It is suggested that the authors analyze the 1608 articles according to their publication date. If the yearly/monthly distribution is not even, there should be time bias in these data. At this time, the author should try to eliminate this impact.

7. PLOS authors have the option to publish the peer review history of their article (what does this mean?). If published, this will include your full peer review and any attached files.

Reviewer #2: No

---

## [Author Response · Author response to Decision Letter 1]

16 Dec 2022

Dear anonymous reviewer!

We would like to thank you for the positive feedback on our article “Correlations between the selection of topics by news media and scientific journals”. We have taken your final comment into account and hope that our explanation and changes are satisfactory so that nothing stands in the way of a publication of our mauscript.

Please find our response to your comment in the document "Response to Reviewers".

---

## [Editor Report · Decision Letter 2]

21 Dec 2022

Correlations between the selection of topics by news media and scientific journals

PONE-D-22-17799R2

Dear Dr. Leidecker-Sandmann,

We’re pleased to inform you that your manuscript has been judged scientifically suitable for publication and will be formally accepted for publication once it meets all outstanding technical requirements.

Kind regards,

Quinn Grundy, PhD, RN

Academic Editor

PLOS ONE
---

## [Editor Report · Acceptance letter]

29 Dec 2022

PONE-D-22-17799R2 

Correlations between the selection of topics by news media and scientific journals 

Dear Dr. Leidecker-Sandmann:

I'm pleased to inform you that your manuscript has been deemed suitable for publication in PLOS ONE. Congratulations! Your manuscript is now with our production department. 

Kind regards, 

on behalf of

Dr. Quinn Grundy 

Academic Editor

PLOS ONE